# Emergence of transmissible SARS-CoV-2 variants with decreased sensitivity to antivirals in immunocompromised patients with persistent infections

Mohammed Nooruzzaman[1,8], Katherine E. E. Johnson[2,8], Ruchi Rani[1], Eli J. Finkelsztein [3], Leonardo C. Caserta [1], Rosy P. Kodiyanplakkal [3], Wei Wang[2], Jingmei Hsu[3,7], Maria T. Salpietro[4], Stephanie Banakis[2], Joshua Albert[2], Lars F. Westblade[5], Claudio Zanettini [5], Luigi Marchionni [5], Rosemary Soave[3], Elodie Ghedin [2,9] ✉, Diego G. Diel [1,9] ✉ & Mirella Salvatore [3,6,9] ✉

We investigated the impact of antiviral treatment on the emergence of SARS-CoV-2 resistance during persistent infections in immunocompromised patients ($n = 15$). All patients received remdesivir and some also received nirmatrelvir-ritonavir ($n = 3$) or therapeutic monoclonal antibodies ($n = 4$). Sequence analysis showed that nine patients carried viruses with mutations in the nsp12 (RNA dependent RNA polymerase), while four had viruses with nsp5 (3C protease) mutations. Infectious SARS-CoV-2 with a double mutation in nsp5 (T169I) and nsp12 (V792I) was recovered from respiratory secretions 77 days after initial COVID-19 diagnosis from a patient sequentially treated with nirmatrelvir-ritonavir and remdesivir. In vitro characterization confirmed its decreased sensitivity to remdesivir and nirmatrelvir, which was overcome by combined antiviral treatment. Studies in golden Syrian hamsters demonstrated efficient transmission to contact animals. This study documents the isolation of SARS-CoV-2 carrying resistance mutations to both nirmatrelvir and remdesivir from a patient and demonstrates its transmissibility in vivo.

Infection with SARS-CoV-2 in immunocompromised patients poses major clinical, therapeutic, and public health challenges. These patients often experience more severe infection outcome(s) than the general population with disease progression being influenced by the treatment of the underlying condition[1]. Moreover, while most people with a competent immune system successfully clear SARS-CoV-2 infection within days, immunocompromised patients may become persistently infected and present prolonged virus replication and shedding. Long-term viral replication contributes to intra-host evolution leading to the emergence of variants with mutations in the virus

---

[1]Department of Population Medicine and Diagnostic Sciences, College of Veterinary Medicine, Cornell University, Ithaca, NY, USA. [2]Systems Genomics Section, NIH/NIAID/DIR/LPD, Bethesda, MD, USA. [3]Department of Medicine, Weill Cornell Medicine, New York, NY, USA. [4]Institutional Biorepository Core, Weill Cornell Medicine, New York, NY, USA. [5]Department of Pathology and Laboratory Medicine, Weill Cornell Medicine, New York, NY, USA. [6]Department of Population Health Science, Weill Cornell Medicine, New York, NY, USA. [7]Present address: Transplantation and Cellular Therapy Program, Perlmutter Cancer Center, NYU Langone Health, New York, NY, USA. [8]These authors contributed equally: Mohammed Nooruzzaman, Katherine E. E. Johnson. [9]These authors jointly supervised this work: Elodie Ghedin, Diego G. Diel, Mirella Salvatore. ✉e-mail: elodie.ghedin@nih.gov; dgdiel@cornell.edu; mis2053@med.cornell.edu

genome[2]. Long-term shedding of SARS-CoV-2 may also favor the spread of these variants, which poses major challenges for disease management and transmission control.

Antiviral treatments with nirmatrelvir-ritonavir [Paxlovid™] that targets the SARS-CoV-2 nsp5 (3C) protease, remdesivir [Veklury™] which inhibits the activity of the RNA-dependent RNA polymerase (RdRp) nsp12, or molnupiravir [Lagevrio™], which induces random and lethal mutations in the viral genome, have been shown to be effective in decreasing viral load and in halting progression to severe disease when administered to high-risk individuals early after infection[3–8]. The effectiveness of these therapeutic interventions is, however, decreased in immunocompromised patients, for which prolonged and repeated treatment courses are often needed, with only modest or no therapeutic benefits. Adding to the low efficacy, antiviral treatment with molnupiravir[9] and possibly remdesivir[10] have been associated with an increase in SARS-CoV-2 genomic diversity, selection of resistance mutations and emergence of novel variants[11]. This can be exacerbated during persistent infection in immunocompromised patients. Although SARS-CoV-2 variants resistant to nirmatrelvir have been generated in vitro[12,13], to date there are limited data on resistant variants recovered from clinical patient samples. Additionally, the impact of nirmatrelvir on the emergence of SARS-CoV-2 resistant variants in immunocompromised patients exhibiting prolonged virus replication remains unknown.

Here, we studied the emergence of SARS-CoV-2 variants in 15 immunocompromised patients that received one or more courses of antiviral therapy over the course of persistent/prolonged SARS-CoV-2 infection. We characterized the effects of select emerging mutations on drug sensitivity in cell culture and assessed the ability of the drug-resistant virus to transmit using a hamster model of SARS-CoV-2 infection.

## Results

The impact of antiviral therapy on the emergence of potentially resistant SARS-CoV-2 variants was investigated. We performed whole genome sequencing of SARS-CoV-2 from nasopharyngeal swab (NPS) samples collected longitudinally from 15 immunocompromised patients (age ranging from 23 to 81 years) with prolonged SARS-CoV-2 infection (duration of 28–190 days) who received one or more courses of remdesivir (average treatment duration 4.2 days; range 2–11 days). Three patients also received a treatment course with nirmatrelvir-ritonavir (Paxlovid™) (Fig. 1a, Table 1, and Supplementary Table 1). All patients were immunocompromised and received chemotherapy for hematological malignancies or (one patient) immunosuppressive therapy following kidney transplant (Table 1).

### Emergence of potential drug resistance mutations

To identify resistant mutations emerging following antiviral therapy, we analyzed nsp5 (target of nirmatrelvir) and nsp12 (target of remdesivir). Nsp5 mutations were identified in 4 of the 15 patients, including one patient who received nirmatrelvir-ritonavir before samples were collected and sequenced and one who was treated with nirmatrelvir-ritonavir following sample collection. In comparison, nsp12 mutations were identified in 9 of the 15 patients (Supplementary Table 2). Most nsp5 and nsp12 mutations were present at low frequencies in the viral sequence data, were detected at single collection time points and did not reach fixation (Supplementary Table 2). However, SARS-CoV-2 sequences recovered from three patients (Patient IDs 11595, 16902, and 17072) had mutations in nsp12 (11595: M794I; 16902: C464Y, C799Y; and 17072: V792I, E796K) that were detected at multiple time points post remdesivir treatment and became the dominant (>50%) variant in the virus population (Fig. 1a, c, d and Supplementary Table 2). Notably, patient 17072, who was treated with nirmatrelvir-ritonavir, carried viruses with nsp5 mutations (T169I and A173T) that were detected at multiple collection time points and present as major

variants (Fig. 1d and Supplementary Table 2), suggesting possible emergence of a multi-drug resistant SARS-CoV-2 variant in this patient. We thus focused our analyses on samples from patients 11595, 16902, and 17072 as they were most likely to be infected with SARS-CoV-2 variants with fixed antiviral-resistant mutations.

Patient 11595 had a history of autoimmune hemolytic anemia on long-term steroids and was admitted to the hospital for the treatment of a newly diagnosed IgM-κ monoclonal gammopathy. The clinical course was complicated by hospital-acquired SARS-CoV-2 for which the patient received remdesivir (day 2–7 post-diagnosis, pd) with clinical improvement; SARS-CoV-2 PCR remained positive. On day 20 pd the patient developed fever and severe hypoxia, requiring high-flow oxygen ventilation. Chest computed tomography (CT) scan showed organizing pneumonia; the patient had no clinical or laboratory findings indicating bacterial infection. Nasal swabs collected on days 14, 20, 26 and 38 pd were positive for SARS-CoV-2 with low Ct values (19.4, 19.25, 20.7, and 20.7, respectively) indicating high viral load and persistent infection. On day 43 pd the clinical scenario worsened and the patient's oxygen requirement rapidly increased. The patient declined invasive procedures and died on day 45 pd. Relevant laboratory data are presented in Supplementary Table 3.

Whole genome sequencing of SARS-CoV-2 directly from NPS collected on days 14, 20, 26 and 38 pd revealed infection with a SARS-CoV-2 20 G (B.1.2) lineage (Fig. 1a and Supplementary Table 1). Single nucleotide variants peaked late in the infection (day 38 pd), 7 days before the patient's death (Fig. 1b). During the disease course, seven SARS-CoV-2 substitutions in the nsp12 emerged (E136A, V166L, Q444K, V792I, M794I, C799F, and V820G) (Fig. 1b and Supplementary Table 2). However, M794I was the only nsp12 substitution to reach consensus levels, initially observed at 17% frequency on day 14 pd and increasing to 90% by day 38 pd in absence of further remdesivir treatment. Additionally, on days 14, 20, and 26 pd, the V792I remdesivir resistance substitution[14,15] was detected at frequencies ranging from 9-31%. Four nsp12 substitutions were identified at single timepoints with frequencies <30%, three of which (E136A, V166L, C799F)[15–18] were detected in the first sequenced sample (day 14 pd) and have been identified in studies focused on the emergence of remdesivir resistant mutations[15–18]. Four mutations were present at multiple timepoints outside the nsp5 and nsp12 coding regions but never became dominant (Supplementary Fig. 1a).

Patient 16902 had high-grade myeloproliferative disease for which they received an allogenic stem cell transplant (SCT). Seventy-eight days after SCT the patient experienced rhinorrhea and fever and was diagnosed with SARS-CoV-2 infection. Chest CT revealed pneumonia leading to hospital admission (day 1–5 pd) for SARS-CoV-2 treatment with Sotrovimab, a monoclonal antibody (mAb) against the SARS-CoV-2 spike protein. The patient was re-admitted to the hospital on day 24 pd with severe hypoxia, fever and diarrhea. Laboratory workup revealed persistent SARS-CoV-2 infection (Ct values: 29.2 on day 24 and 25.1 on day 28), leukopenia and pneumonia. The patient received empiric antibiotic therapy (day 24–26 pd) and COVID-19-specific treatment with remdesivir (days 26–37 pd), dexamethasone (days 26–53 pd), Tocilizumab (a mAb against interleukin-6 receptor, day 31 pd) and high-flow oxygen supplementation until day 45 pd when they developed rapidly progressing acute hypoxemic respiratory failure. On day 47 pd SARS-CoV-2 Ct value was 19.1. A second cycle of remdesivir was initiated (days 47–51 pd) with no improvement. On day 53 the patient opted for non-invasive treatments and died on day 54 pd (Supplementary Table 4).

Whole SARS-CoV-2 genome sequencing from NPS collected on days 1, 7, 11, 14, 26, 28, 36, 42 and 47 pd revealed infection with a 21 K (BA.1.15) lineage virus. The number of single nucleotide variants identified in SARS-CoV-2 sequences was highest on day 36 pd (day 10 of the first course of remdesivir treatment) (Fig. 1c). Two nsp12 substitutions, C464Y and C799Y, were detected at frequencies of 98 and

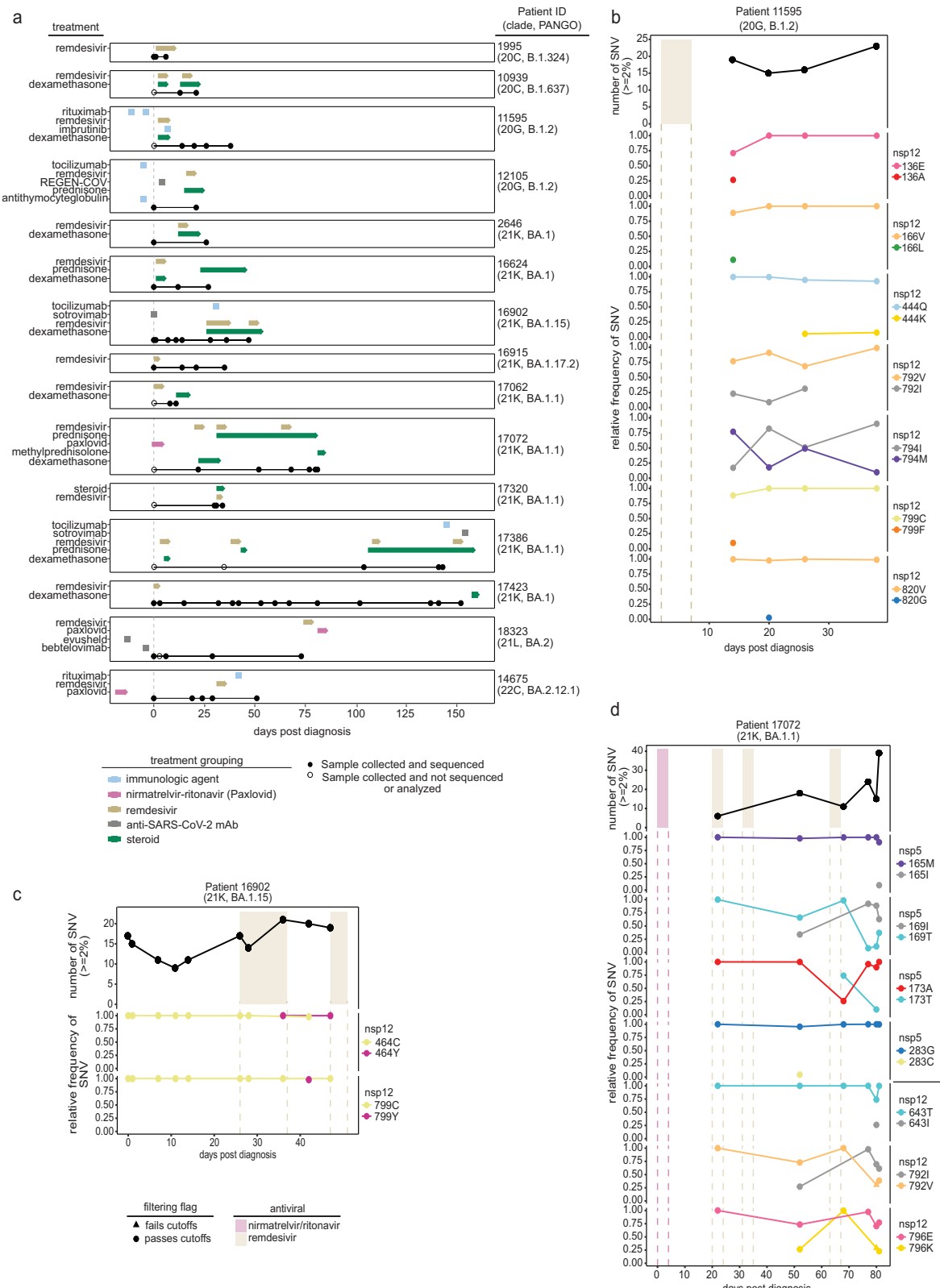

100% following remdesivir treatment. Nsp12 mutation C464Y first emerged at 99.9% on day 36 pd before dropping below our limit of detection (0-1.99%) on day 42 and then increasing to 99.9% frequency on day 47, coinciding with the start date of a second course of remdesivir. An N-terminal domain spike mutation, known to impact antibody binding affinity (T76I), was also present at nearly 100% on days 36 and 42 pd (Supplementary Fig. 1b)[19]. The nsp12 mutation

C799Y was only observed on day 42 pd but was present at 98% frequency. Though this substitution differs from the one observed in patient 11595 (C799F), it highlights the importance of residue 799 in mediating remdesivir resistance. The virus sequenced from this patient also carried the Spike E340D mutation, previously associated with resistance to Sotrovimab (Supplementary Fig. 1b)[20–23]. This mutation was present at 100% frequency on day 7 pd and remained in

**Fig. 1 | Emergence of SARS-CoV-2 variants in immunocompromised patients.**
**a** The treatment course and sample collection from 15 patients with persistent
SARS-CoV-2 infections. Clade and PANGO lineage designations are included with
Patient IDs. All time points along the *x* axis are referenced from the date of each
patient's initial positive COVID-19 test result, with day 0 marking the date of the
first positive test at the New York Presbyterian (NYP) Hospital (dashed gray line).
Treatment courses are specified along the *y* axis for each patient and colored
based on the treatment category. Nasopharyngeal swab samples successfully
sequenced and used for analyses are represented as black points along the solid
black line for each patient. The total number of unique single-nucleotide variants
(SNVs) found ≥2% across the genome (top) over the course of the infection in
patients **b** 11595, **c** 16902, and **d** 17072 compared to their treatment courses with
nirmatrelvir-ritonavir (pink) or remdesivir (tan). SNVs are determined by com-
paring each sample to their respective clade consensus sequences ("Methods").
The relative frequency of non-synonymous SARS-CoV-2 SNVs in nsp5 and nsp12
coding regions during infection are shown below. Variant data are grouped by the
coding region, amino acid position, and amino acid residue, with the color
representing the amino acid and the shape indicating whether the variant was
found above (passes, circle) or below (fails, triangle) our detection cutoffs
("Methods"). All time points along the *x* axis are referenced from the date of each
patient's initial positive COVID-19 test result, with day 0 marking the date of the
first positive test at the NYP hospital.

the following collections indicating that the nsp12 mutations C464Y
and C799Y associated with remdesivir resistance arose on a virus
already resistant to Sotrovimab.

Patient 17072 was affected by stage III IgG multiple myeloma that
had been treated with Elranatamab, a B-cell maturation antigen - CD3
T-bispecific antibody. The patient experienced a mild cough and was
diagnosed with COVID-19 and treated as an outpatient with a 5-day
course of nirmatrelvir/ritonavir with symptom improvement. On day
18 pd, the patient developed severe cough and dyspnea and presented
to an emergency room where was diagnosed with pneumonia and
empirically treated with antibiotics. Two days later (day 20 pd) the
symptoms worsened, and the patient developed fever and hypoxia
requiring oxygen supplementation, leading to hospital admission. PCR
testing on NPS continued to be positive for SARS-CoV-2. The patient
was treated with remdesivir (days 20–24 pd), dexamethasone and
antibiotics until symptoms improved and was discharged home. One
week later the patient was re-admitted to the hospital due to recurrent
fever, cough, and hypoxia. SARS-CoV-2 PCR remained positive, but the
microbiology work-up was otherwise negative. Chest imaging showed
pulmonary fibrosis and organizing pneumonia. During hospitalization
the patient received two additional courses of remdesivir (day 30–35
and 63–67 pd) due to persistent SARS-CoV-2 detection, fluctuating
oxygen requirement and fever. On day 80 of hospitalization, the
patient's oxygen requirement rapidly escalated, they developed acute
hypoxemic respiratory failure requiring intubation and multisystem
organ failure, and died on day 85 pd. Details of laboratory tests and
viral loads during the infection course are shown in Supplementary
Table 5.

Whole genome sequencing on NPS collected on days 22, 52, 68,
77, 80 and 81 pd revealed infection with a 21 K (BA.1.1) lineage virus.
The number of single nucleotide variants increased throughout the
infection, peaking after the third course of remdesivir treatment on
day 81 pd (Fig. 1d). A total of seven non-synonymous substitutions
were identified in nsp5 (M165I, T169I, A173T, and G283C) and nsp12
(T643I, V792I and E796K) coding regions during infection (Supple-
mentary Table 2). Three of these substitutions (nsp5: M165I, G283C,
and nsp12: T643I) were identified at single time-points and at relatively
low frequencies <50% (Fig. 1d). The remaining four mutations (nsp5:
T169I, A173T, and nsp12: V792I[15], E796K) reached consensus levels
(≥50%) following the third course of remdesivir treatment. Nine addi-
tional amino acid substitutions (nsp3: S454R, R568C, V573I, N1369S,
nsp4: T295I, membrane protein: I76T, A188S, nucleocapsid: A155D,
D402H) reached >50% following the third remdesivir treatment (Sup-
plementary Fig. 1c). Two of the mutations (nsp3: N1369S and nsp4:
T295I) shared similar frequency dynamics as nsp5: T169I and nsp12:
V792I, suggesting the four mutations may be linked (Supplemen-
tary Fig. 1c).

### Recovery of infectious SARS-CoV-2 carrying putative antiviral-resistant mutations in nsp5 and nsp12

To study the effect of identified mutations on SARS-CoV-2 antiviral
resistance, we performed virus isolation in Vero E6 TMPRSS2 cells
using clinical samples from patients 11595, 16902, and 17072. The

SARS-CoV-2 (21-CoV-1759α, B.1.2, lineage 20 G) virus recovered from
patient 11595 was isolated from a sample collected on day 38 pd
(Supplementary Fig. 2a and Supplementary Table 6). The consensus
sequence obtained from the clinical sample collected on day 38 pd
from patient 11595 contained the nsp12 mutation M794I, which has
been associated with remdesivir resistance[15]. Sequencing of the SARS-
CoV-2 isolate, however, revealed that this mutation was not stable
upon virus replication in cell culture as it reverted to the prototypic
B.1.2 lineage nsp12 residue (M794) (Supplementary Table 6). Infectious
SARS-CoV-2 (22-CoV-888α) was only recovered from the day 0 sample
from patient 16902 (the day of initial COVID-19 diagnosis) (Supple-
mentary Fig. 2a and Supplementary Table 6). At this stage of infection,
no nsp12 mutations were detected (Supplementary Table 6).

Infectious SARS-CoV-2 (BA.1.1, 21 K lineage) was isolated from
patient 17072, who received nirmatrelvir/ritonavir and remdesivir
treatment courses, from samples collected on days 77 and 81 pd
(Fig. 1a and Supplementary Table 6). The SARS-CoV-2 sequences
recovered from both clinical samples contained a nsp5 T169I mutation,
and two nsp12 mutations, V223I and V792I, when compared to a pro-
totypic BA.1.1 virus sequence (SARS-CoV-2 isolate NYC3/18-22).
Sequencing confirmed that both virus isolates retained the nsp5
(T169I) and nsp12 (V792I) mutations detected in the clinical samples
upon three serial passages in cell culture. The isolate recovered from
day 81 pd (22-CoV-1694α), however, lost the nsp12 mutation V223I
(Supplementary Table 6), likely due to cell culture adaptation. We
focused on the characterization of SARS-CoV-2 isolate NYC3/18-22
containing two putative antiviral-resistant mutations in nsp5 (T169I)
and nsp12 (V792I).

### Replication properties of SARS-CoV-2-nsp5$^{T169I}$nsp12$^{V792I}$ virus

The replication kinetics and plaque size and morphology of SARS-CoV-
2-nsp5$^{T169I}$nsp12$^{V792I}$ (BA.1.1) were investigated and compared to a pro-
totype SARS-CoV-2 BA1.1 strain NYI45-21 (wild-type, WT) in vitro. While
similar replication kinetics were observed for both viruses in Vero E6
and Vero E6 TMPRRS2 cells (Supplementary Fig. 2b), the SARS-CoV-2-
nsp5$^{T169I}$nsp12$^{V792I}$ isolate presented higher replication at early times
post-infection (pi) as evidenced by higher viral yields at 8 ($P \leq 0.05$)
and 12 h ($P \leq 0.01$) post-infection (hpi). Additionally, the SARS-CoV-2-
nsp5$^{T169I}$nsp12$^{V792I}$ isolate produced smaller plaques in Vero E6
(2.14 mm$^2$ vs 2.99 mm$^2$, $P \leq 0.01$) and Vero E6 TMPRSS2 cells (6.1 mm$^2$
vs 8.3 mm$^2$, $P \leq 0.0001$) when compared to the WT virus (Supple-
mentary Fig. 2c, d).

### The SARS-CoV-2-nsp5$^{T169I}$nsp12$^{V792I}$ variant presents decreased sensitivity to antiviral therapies

The sensitivity of the SARS-CoV-2-nsp5$^{T169I}$nsp12$^{V792I}$ variant to nirma-
trelvir or remdesivir was assessed and compared to that of WT virus
in vitro. We used three treatment conditions: 1. Pre-treatment of the
cells with either antiviral drug (4 h) before viral infection (pre-treat-
ment), 2. Treatment of the cells at the time of infection (simultaneous
treatment), and 3. Treatment after virus infection (post-treatment)
(Fig. 2a). Treatment with nirmatrelvir led to complete inhibition of WT
virus replication at a concentration of 3.13 μM during the pre- and

**Table 1 | Clinical characteristics, immunosuppressive treatments and COVID-19 course**

| ID | Age (range) | Comorbidities | Fludarabine/ melphalan/ TBI 400 cGy | Alemtuzumab | Tacrolimus | MMF | Anti-thymocyte globulin | Rituximab | Steroids | Other | Previous covid vaccine | SARS-CoV2 anti-S Ab | SARS-CoV2 anti-N Ab | Days PCR + | Neutropenia (<1.42×10⁹/ml) | Lymphopenia (<1.17×10⁹/ml) | Total IgG 610–1616 mg/dL | PNA | ICU | O₂ | RDV | DEXA | Toci | mAb | PAX | Covid outcome |
|---|---|---|---|---|---|---|---|---|---|---|---|---|---|---|---|---|---|---|---|---|---|---|---|---|---|---|
| 1995 | 60–65 | MDS, SCT | Yes | Yes | Yes | | | | Yes | Ruxolitinib | No | No | Yes | 28 | No | Yes | 621 | Yes | No | No | Yes | Yes | No | No | No | Discharged home |
| 2646 | 40–45 | AML | Yes | Yes | | | | | | Decitabine Venetoclax | No | No | Yes | 32+ | No | No | 695 | Yes | Yes | HFNC | Yes | Yes | No | No | No | Discharged home. |
| 10939 | 80–85 | Kidney Tx | | | Yes | Yes | | | | | No | No | Yes | 33' | No | Yes | 1072 | Yes | Yes | INV | Yes (x2) | Yes | No | No | No | Deceased |
| 11595 | 66–70 | AIHA, MGUS | | | | | | Yes | Yes | Ibrutinib | No | No | No | 38 | Yes | Yes | 217 | Yes | No | HFNC | Yes | Yes | No | No | No | Deceased |
| 12105 | 66–70 | T-PLL, SCT | Yes | Yes | Yes | Yes | | Yes | Yes | PLEX | No | No | No | 58' | Yes | Yes | 232 | Yes | Yes | INV | Yes | Yes | Yes | Casi/indi | No | Deceased |
| 14675 | 66–70 | CLL | | | | | | Yes | Yes | Polartuzumab Etoposide Doxorubicin CFX | Yes | No | No | 66 | No | Yes | 578 | Yes | No | No | No | No | No | Tixa/Cilga | Yes | Discharged home |
| 16624 | 60–65 | Mantle cell lymphoma, SCT | | | Yes | | | Yes | Yes | | Yes | Yes | No | 42 | No | Yes | 418 | Yes | No | NC | Yes | Yes | No | No | No | Discharged home |
| 16902 | 70–75 | MDS, SCT | Yes | Yes | Yes | | | Yes | | | No | Yes | No | 50' | Yes | Yes | 484 | Yes | No | INV | Yes (x2) | Yes | Yes | Sotrovimab | No | Deceased |
| 16915 | 20–25 | AML, SCT | Yes | Yes | Yes | Yes | | Yes | | | No | No | No | 63 | No | No | 563 | Yes | No | No | Yes | No | No | Tixa/Cilga | No | Discharged home |
| 17062 | 26–30 | BMFS, SCT | Yes | | Yes | Yes | Yes | | | Fludarabine, CFX | No | Yes | Yes | 72 | Yes | Yes | 270 | Yes | Yes | NC | Yes | Yes | No | No | No | Deceased |
| 17320 | 70–75 | Mantle cell lymphoma | | | | | | | | Fludarabine, CFX CAR-T cells | Yes | No | No | 34 | Yes | Yes | 447 | Yes | No | No | Yes | Yes | No | No | No | Discharged home |
| 17386 | 60–65 | DLBCL | | | | | | Yes | Yes | CHOP | No | Yes | No | 157 | No | Yes | 810 | Yes | No | NC | Yes (x4) | Yes | Yes | Sotrovimab | No | Discharged home |
| 17423 | 60–65 | B-ALL, SCT, GVHD | Yes | | Yes | Yes | | | | Ruxolitinib | Yes | Yes | No | 190 | No | Yes | 797 | Yes | No | No | Yes | Yes | No | No | No | Deceased |
| 18323 | 36–70 | AML, SCT | Yes | | Yes | Yes | | | | Desatinib, | No | No | No | 131 | No | Yes | 926 | Yes | No | No | No | No | No | Tixa/Cilga Bebtel | Yes | Discharged home |
| 17072 | 60–65 | MM | | | | | | | Yes | Eltranatam | Yes | No | No | 81 | No | Yes | 441 | Yes | Yes | IMV | Yes (x3) | Yes | No | No | Yes | Deceased |

*AIHA* autoimmune hemolytic anemia, *ALL* acute lymphoid leukemia, *AML* acute myeloid leukemia, *aSCT* autologous stem cell transplant, *BMFS* bone marrow failure syndrome, *CHOP* cyclophosphamide, doxorubicin, vincristine, prednisone, *CLL* chronic Lymphoid Leukemia, *DLBCL* diffuse large B cell Lymphoma, *GVHD* graft versus host disease, *MDS* myelodysplastic syndrome, *MGUS* monoclonal gammopathy of undetermined significance, *MM* multiple myeloma, *MGUS* monoclonal gammopathy of undetermined significance, *Bebtel* Bebtelovimab, *Casi/indi* casirivimab and imdevimab, *DEXA* dexamethasone, *HFNC* high-flow nasal cannula, *ICU* intensive care unit, *IMV* invasive mechanical ventilation, *NC* nasal cannula, *NIV* non-invasive ventilation, *O₂* oxygen requirements, *PAX* nirmatrelvir/ritonavir (Paxlovid™), *PNA* pneumonia, *RDV* remdesivir, *SCT* stem cell transplant, *TBI* total body irradiation, *T-PLL* T-cell prolymphocytic leukemia, *Tx* transplant, *CFX* cyclophosphamide, *Toci* Tocilizumab, *Tixa/Cilga* Tixagevimab/Cilgavimab (Evusheld™).

'indicates that the patient died while SARS-CoV2 positive; *indicates that the patient died in the following month after hospitalization without retesting for SARS-CoV2.

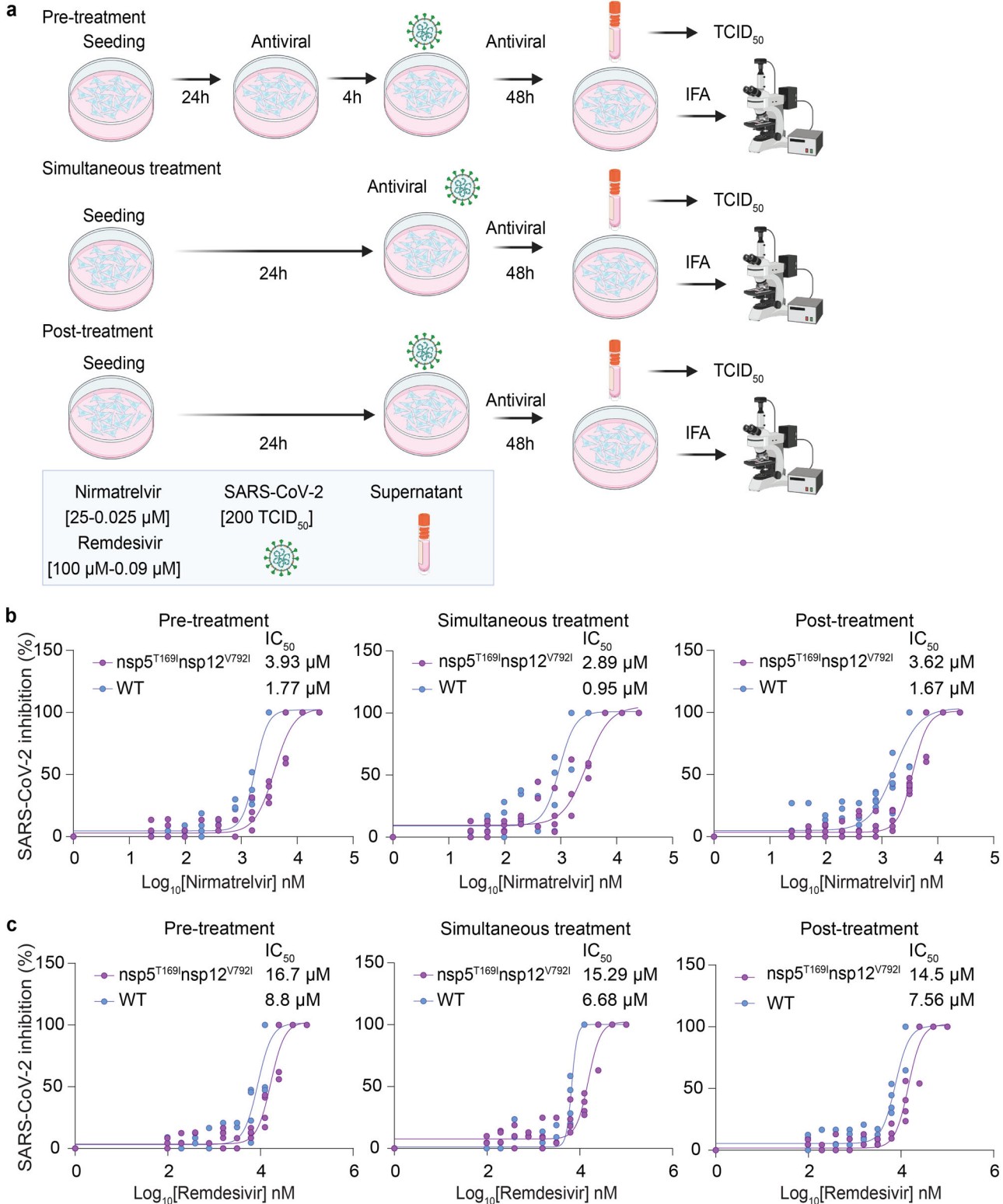

**Fig. 2 | Sensitivity of SARS-CoV-2-nsp5$^{T169I}$nsp12$^{V792I}$ virus to nirmatrelvir and remdesivir in vitro. a** Experimental layout showing the treatment plans to study the sensitivity of the SARS-CoV-2-nsp5$^{T169I}$nsp12$^{V792I}$ virus to antiviral therapies. **b** Nirmatrelvir resistance of SARS-CoV-2. Vero E6 cells were treated with increasing doses of nirmatrelvir (0.05–25 μM) following three treatment conditions (pre-, simultaneous, and post-treatment) and infected with 200 TCID$_{50}$/well of SARS-CoV-2-nsp5$^{T169I}$nsp12$^{V792I}$ and WT viruses. Virus titers in supernatant of treated/infected cells was quantified by limiting dilution and the percent inhibition value was calculated to obtain the 50% inhibitory concentration (IC$_{50}$). Results represent the average

of 4 (pre- and simultaneous treatment) and 6 (post-treatment) replicates and three independent experiments. **c** Remdesivir resistance of SARS-CoV-2. Vero E6 cells were treated with increasing doses (0.09–100 μM) of remdesivir following three treatment regimens and infected with 200 TCID$_{50}$/well of SARS-CoV-2-nsp5$^{T169I}$nsp12$^{V792I}$ and WT isolates. Virus titers in supernatants were quantified by limiting dilution and the percent inhibition value was calculated to obtain the 50% inhibitory concentration (IC$_{50}$). Results represent the average of four replicates from two independent experiments. **a** Created with BioRender.com released under a Creative Commons Attribution-NonCommercial-NoDerivs 4.0 International license.

simultaneous treatment conditions, while 6.26 μM of nirmatrelvir were required for complete inhibition of WT replication during the post-treatment condition (Supplementary Fig. 3). Decreased nirmatrelvir sensitivity was observed for the SARS-CoV-2-nsp5$^{T169I}$nsp12$^{V792I}$ virus. Complete inhibition of SARS-CoV-2-nsp5$^{T169I}$nsp12$^{V792I}$ virus replication was observed with a dose of 6.26 μM of nirmatrelvir (2-fold higher than for WT virus) in pre- and post-treatment experiments, while 12.5 μM of nirmatrelvir were required for complete inhibition of SARS-CoV-2-nsp5$^{T169I}$nsp12$^{V792I}$ virus replication during simultaneous treatment (Supplementary Fig. 3). The IC$_{50}$ calculations based on viral titers in the supernatant of infected and treated cells corroborated these findings. The IC$_{50}$ for nirmatrelvir against the WT virus were calculated at 1.77, 0.95 and 1.67 μM during pre-, simultaneous- and post-treatment studies, respectively. The IC$_{50}$ for nirmatrelvir against the SARS-CoV-2-nsp5$^{T169I}$nsp12$^{V792I}$ virus were 3.93 μM (2.22-fold higher than for WT), 2.89 μM (3.04-fold higher), and 3.62 μM (2.17-fold higher) in pre-, simultaneous and post-treatment studies, respectively (Fig. 2b and Supplementary Table 7).

Treatment with remdesivir led to similar results, with the SARS-CoV-2-nsp5$^{T169I}$nsp12$^{V792I}$ virus presenting decreased sensitivity to the drug when compared to the WT virus. Remdesivir treatment led to complete inhibition of WT virus replication with a dose of 25 μM during pre- and simultaneous treatment and with a dose of 50 μM during post-treatment, while 100 μM of remdesivir were required for complete inhibition of SARS-CoV-2-nsp5$^{T169I}$nsp12$^{V792I}$ virus replication in all treatment conditions (Supplementary Fig. 3). For the WT virus, the IC$_{50}$ of remdesivir was 8.8, 6.68 and 7.56 μM, while for the SARS-CoV-2-nsp5$^{T169I}$nsp12$^{V792I}$ virus the IC$_{50}$ was 16.7 (1.9-fold higher than WT), 15.29 (2.34-fold higher) and 14.5 (1.95-fold higher) μM for pre-, simultaneous-, and post-treatment, respectively (Fig. 2c and Supplementary Table 7). These results confirm that the SARS-CoV-2-nsp5$^{T169I}$nsp12$^{V792I}$ virus presents lower sensitivity to nirmatrelvir and remdesivir in cell culture in vitro.

## Combination treatment with nirmatrelvir and remdesivir inhibited replication of SARS-CoV-2-nsp5$^{T169I}$nsp12$^{V792I}$

We next evaluated the antiviral efficacy of combination nirmatrelvir and remdesivir treatment in vitro. Each drug was tested individually or in combination at 0.25×, 0.5×, 1×, 2× and 4× their IC$_{50}$ against SARS-CoV-2-nsp5$^{T169I}$nsp12$^{V792I}$ following the three treatment conditions described above (pre-, simultaneous-, and post-treatment). Treatment with nirmatrelvir or remdesivir alone resulted in inhibition of virus replication when the drugs were used at 2× their IC$_{50}$, whereas combination treatment with nirmatrelvir and remdesivir resulted in inhibition of virus replication at 1× IC$_{50}$ (Fig. 3a–c), indicating additive effects of combination treatment with both antivirals (Fig. 3a–c). Quantification of virus release from treated cells supported these findings, as evidenced by reduced viral titers in the supernatant of infected cells in the combination treatment when 1× and 0.5× IC$_{50}$ doses were used when compared to the treatments with each drug alone (Fig. 3d).

## Structural analysis of nsp5 T169I and nsp12 V792I substitutions

We next explored the effect of nsp5 T169I and nsp12 V792I substitutions on protein structure and interaction with nirmatrelvir and remdesivir. Structural alignment of nsp5 T169I and nsp12 V792I models with their respective crystal structure revealed a significant degree of structural similarity with root mean squared deviation (RMSD) values of 0.05 Å and 0.09 Å, respectively (Fig. 4a, b).

A docking analysis was performed to investigate the interactions of nsp5 T169I and nsp12 V792I with nirmatrelvir and remdesivir (as remdesivir triphosphate, RTP) drug molecules, respectively. The binding energy of nirmatrelvir with nsp5 T169I was lower (−6.9 kcal/mol vs −7.8 kcal/mol) and interacting residues in the binding pocket differed when compared to the WT nsp5 protein (Fig. 4c, d and

Supplementary Table 8). In nsp12, the binding energy between mutant nsp12 V792I and WT proteins and RPT were identical (−7.1 kcal/mol); however, major differences in interacting residues, hydrogen bonds and hydrophobic interactions were observed (Fig. 4e, f and Supplementary Table 9). In the WT nsp12, ten hydrogen bonds with RTP, involving a total of eight amino acids (including two residues, Ser759 and Asp760, from the main catalytic SDD motif) are predicted, while the mutant nsp12 V792I is predicted to form only five hydrogen bonds with RTP involving five amino acid residues (Fig. 4e, f and Supplementary Table 9). The number of hydrophobic interactions between nsp12 and RTP increased from five in the WT to nine in the mutant nsp12 V792I protein (Fig. 4e, f and Supplementary Table 9). These structural changes in nsp5 T169I and nsp12 V792I may decrease the ability of the drugs to interact effectively with the catalytic motif of the targeted proteins.

## The SARS-CoV-2-nsp5$^{T169I}$nsp12$^{V792I}$ virus efficiently transmits in a hamster model of infection

The transmissibility of the SARS-CoV-2-nsp5$^{T169I}$nsp12$^{V792I}$ isolate was assessed in golden Syrian hamsters (Fig. 5a). All inoculated and contact animals had subclinical disease and gained weight throughout the 14-day experimental period (Fig. 5b). SARS-CoV-2 RNA was detected by RT-PCR in oropharyngeal swab (OPS) samples of all inoculated and contact animals between days 1 and 10 post-inoculation/contact (Fig. 5c), indicating efficient transmission of both SARS-CoV-2-nsp5$^{T169I}$nsp12$^{V792I}$ and WT viruses to contact animals. Infectious SARS-CoV-2 was detected mainly between days 1 and 2 pi with sporadic detection at later times on days 3-6 pi (titers ranging between 1.05 and 3.3 log TCID$_{50}$.mL$^{−1}$). In contact animals, infectious virus was first detected on days 3 and 4 post-contact in SARS-CoV-2-nsp5$^{T169I}$nsp12$^{V792I}$ and WT virus groups, respectively (Fig. 5d, e). Viral load assessment in tissues on day 14 showed that nasal turbinate of all inoculated animals from both virus groups presented relatively high levels of viral RNA, while three out of four contact animals in each group tested positive (Fig. 5f). Low levels of viral RNA were detected in trachea and lungs of inoculated animals, whereas viral RNA was only occasionally detected in these tissue sites of contact animals. No infectious virus was recovered from any of the tissues analyzed. Whole genome sequencing of SARS-CoV-2 from OPS samples collected between days 1 and 7 from SARS-CoV-2-nsp5$^{T169I}$nsp12$^{V792I}$ inoculated and contact hamsters showed that both nsp5$^{T169I}$ and nsp12$^{V792I}$ mutations were maintained in the virus upon replication and transmission in hamsters. Virus neutralization assays performed in serum samples from inoculated and contact animals confirmed seroconversion of all inoculated and 3/4 contact animals in each group (Fig. 5g). These results demonstrate efficient infection, replication, and transmission of the SARS-CoV-2-nsp5$^{T169I}$nsp12$^{V792I}$ virus.

## Discussion

The COVID-19 pandemic has underscored the complex challenges involved in managing the disease in immunocompromised individuals. By examining the infection dynamics and genomic evolution of SARS-CoV-2 in immunocompromised patients with prolonged or persistent viral infections who received antiviral treatments (remdesivir, and in some cases, nirmatrelvir-ritonavir), we discovered that nsp12 mutations commonly emerged in 60% of subjects (9 out of 15). This proportion is higher than what was observed in patients participating in the Remdesivir Phase 3 Adaptive COVID-19 Treatment Trial-1 (ACTT-1) which reported about 40% of the patients had single mutations with no differences between patients treated with remdesivir or placebo[24]. Here SARS-CoV-2 from seven of the study patients carried mutations that were detected in more than one sample over time, with nsp12 V792I present in multiple samples for two of the patients (11595 and 17072). The nsp12 V792I mutation was originally described in vitro

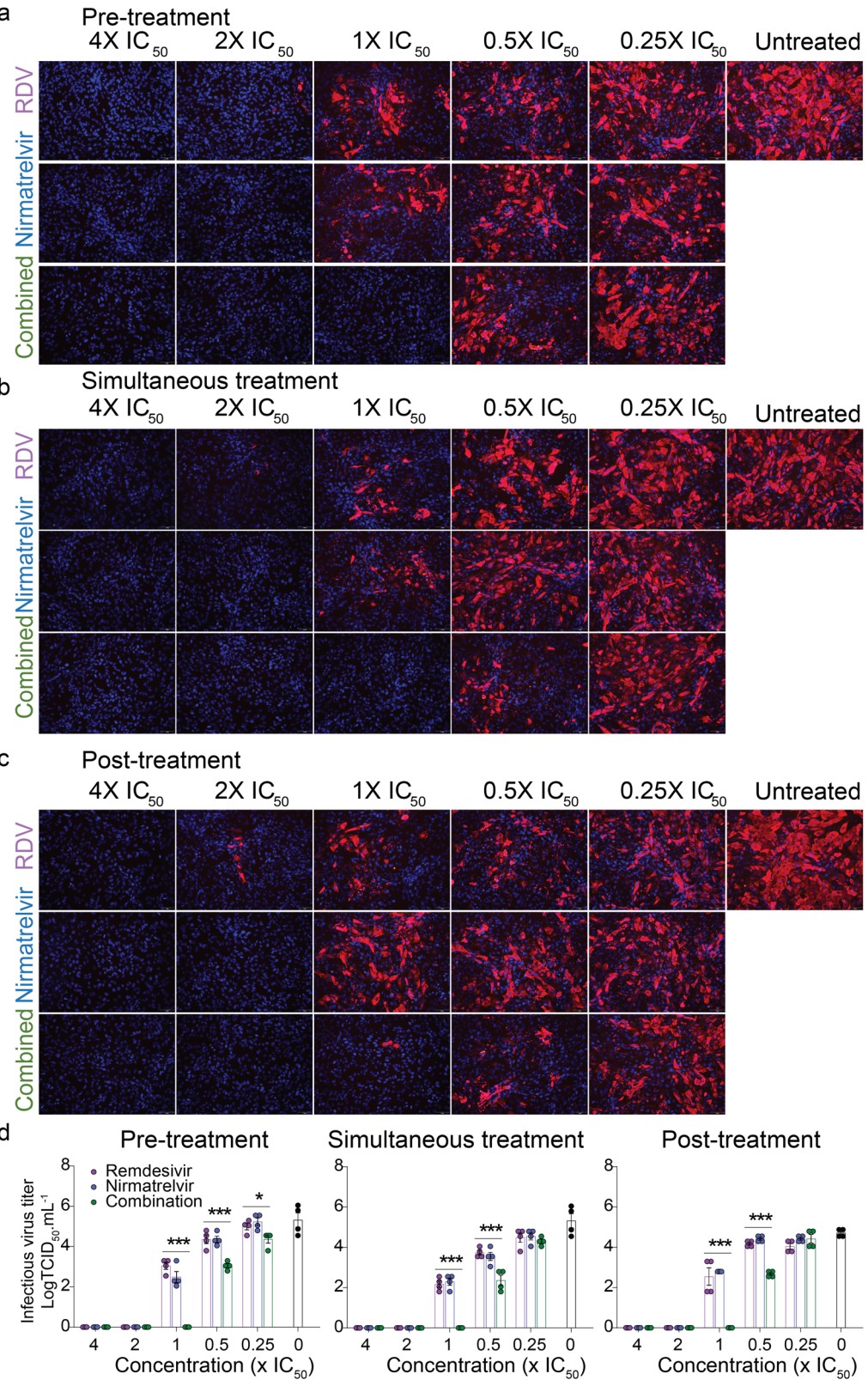

**Fig. 3 | Efficacy of combination nirmatrelvir and remdesivir therapy against SARS-CoV-2-nsp5$^{T169I}$nsp12$^{V792I}$ virus in vitro.** Vero E6 cells were treated with indicated concentration of nirmatrelvir or remdesivir alone or in combination at pre-infection (**a**), at the time of infection (**b**) or at post-infection (**c**) and infected with 200 TCID$_{50}$/well of SARS-CoV-2-nsp5$^{T169I}$nsp12$^{V792I}$. After 48 h, the cell mono-layer was fixed and stained with SARS-CoV-2 N specific monoclonal antibody (red) and the nucleus was counterstained with DAPI (blue). White Bar 1 mm. **d** Virus titration (TCID$_{50}$.mL$^{-1}$) in the supernatant of nirmatrelvir, remdesivir or combination treatment from (**a**–**c**). The results represent mean ± SEM of four replicates from two independent experiments. Two-way ANOVA with Bonferroni multiple comparison test, *$P < 0.05$ and ***$P < 0.001$.

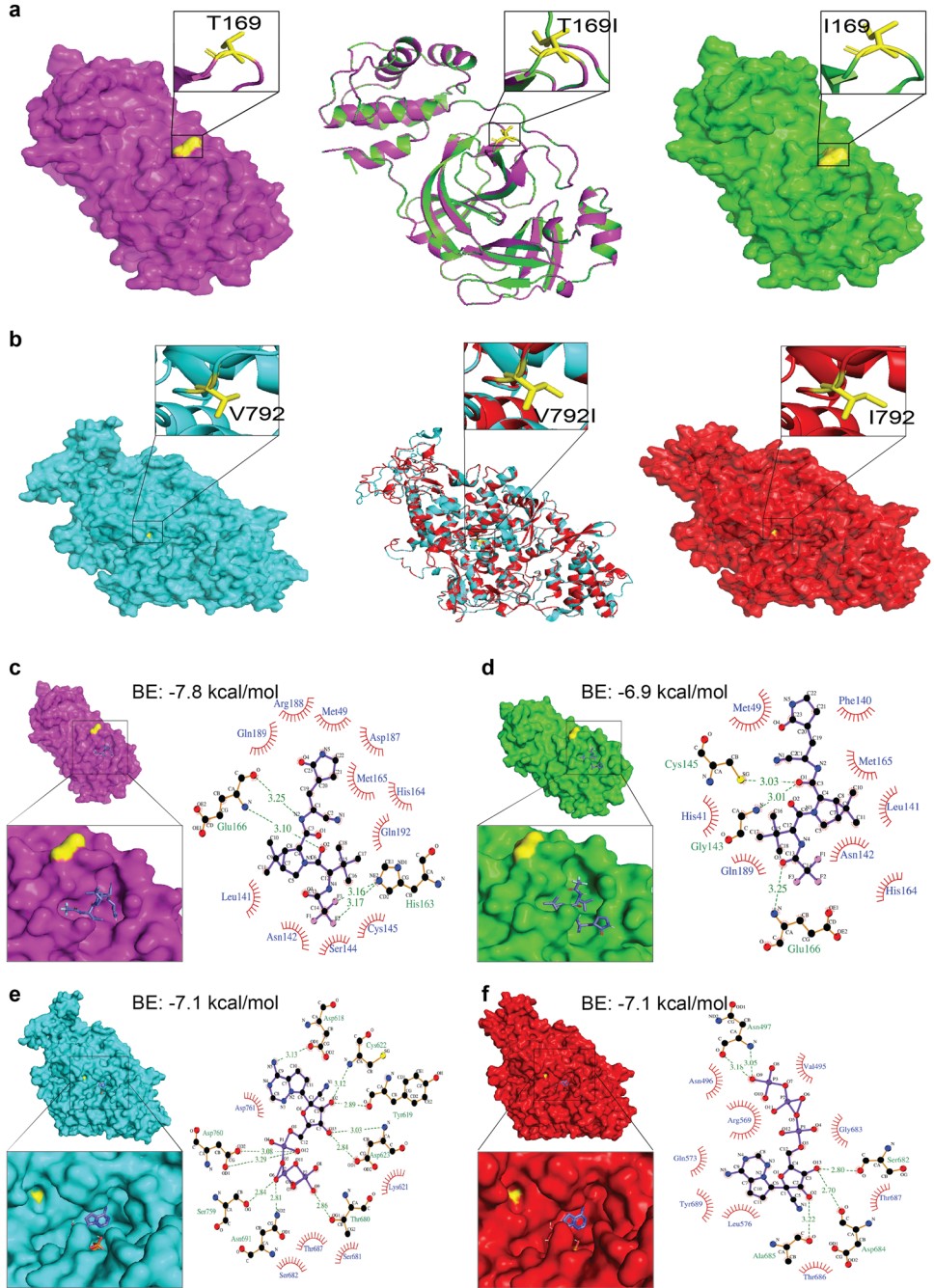

**Fig. 4 | Structural analysis of nsp5 T169I and nsp12 V792I substitutions.** Comparison of nsp5 (**a**) and nsp12 (**b**) protein models between WT (NYI45-21) and mutant nsp5[T169I] and nsp12[V792I] proteins. **a, b**, left panel: WT: Surface representation of nsp5 (magenta) and nsp12 (cyan) proteins with specific amino acid substitutions highlighted in the inset for each protein. The position of substituted amino acids is depicted in yellow sticks with proper labelling. **a, b**, middle panel: Superimposed Image: Cartoon representation of the superimposed image of WT (magenta/cyan) and nsp5[T169I] and nsp12[V792I] proteins (green/red), depicting specific substitutions in the inset box. **a, b**, right panel: nsp5[T169I] and nsp12[V792I]: nsp5 (magenta) and nsp12 (cyan) proteins with specific amino acid substitutions highlighted in inset for each protein in box. The position of the substituted amino acids is shown in yellow sticks with proper labelling. Note: The protein models of the nsp5[T169I] and nsp12[V792I] proteins was generated using Swiss MODELLER, resulting in RMSD values of 0.003 Å for nsp5 and 0.026 Å for nsp12 proteins, indicating high structural

similarity. Surface representation of nsp5 proteins of (**c**) WT and (**d**) nsp5[T169I] protein docked to nirmatrelvir drug molecules shown in the inset using PyMOL analysis tool. 2D depiction of nsp5 protein residues interactions with nirmatrelvir drug molecules via LIGPLOT[+] tool where hydrogen bonds are shown in green dotted lines with a distance in Ångström (Å) and hydrophobic interactions are shown in curvature. Surface representation of nsp12 proteins of **e** WT and **f** mutant nsp12[V792I] docked remdesivir drug molecules shown in inset for each using PyMOL tool. 2D depiction of nsp12 protein residues interactions with remdesivir drug molecule via LIGPLOT[+] tool where hydrogen bond are shown in green dotted lines with a distance in Å and hydrophobic interactions are shown in curvature. The colour scheme is consistent throughout the figures, with magenta representing nsp5 and cyan representing nsp12 of WT isolate, and green representing nsp5 and red representing nsp12 of nsp5[T169I]nsp12[V792I] isolate with substituted amino acid highlighted in yellow. Note: BE Binding energy.

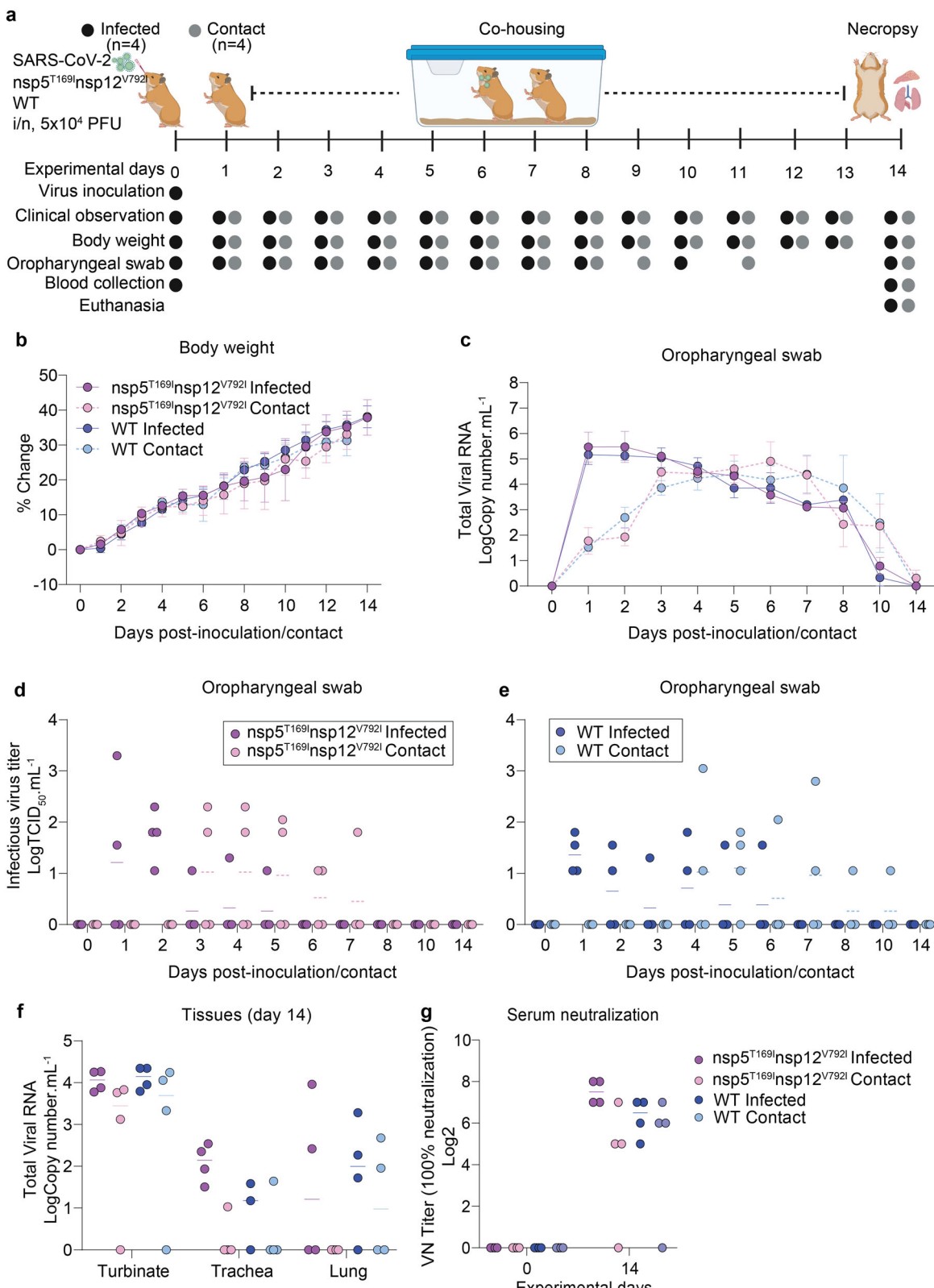

**Fig. 5 | The SARS-CoV-2-nsp5$^{T169I}$nsp12$^{V792I}$ virus efficiently transmitted to contact golden Syrian hamsters. a** Experimental design. **b** Changes in body weight of hamsters following intranasal inoculation of SARS-CoV-2-nsp5$^{T169I}$nsp12$^{V792I}$ and WT viruses and in contact animals throughout the 14-day experimental period. **c** SARS-CoV-2 RNA load in oropharyngeal swabs quantified by rRT-PCR. (**b-c**) Data represents mean ± SEM of four animals/group/timepoint. **d, e** Infectious SARS-CoV-2 loads in oropharyngeal swabs determined using endpoint dilutions and expressed as TCID$_{50}$.mL$^{-1}$. **f** SARS-CoV-2 RNA load in nasal turbinate, trachea and lungs quantified by rRT-PCR. **g** Neutralizing antibody responses to SARS-CoV-2 assessed by virus neutralization assay (100% neutralization) in serum. **d–g** Data represents median of four animals/group/timepoint. **a** Created with BioRender.com released under a Creative Commons Attribution-NonCommercial-NoDerivs 4.0 International license.

after remdesivir treatment in cell culture and was associated with a -2.6-fold decreased sensitivity to remdesivir[15]. This mutation emerged in 7.14% of SARS-CoV-2 genomes sequenced from immunocompromised participants that were included in the large COG-UK dataset, while it was infrequent in the general population[25]. These differences in emergence and persistence of certain mutations after remdesivir therapy support the importance of including immunocompromised populations in clinical trials leading to drug licensing.

Mutations in the nsp5 protein were found in SARS-CoV-2 sequences recovered from four patients in our study, two of which did not receive nirmatrelvir-ritonavir, confirming that naturally occurring mutations in nsp5 in untreated individuals can potentially lead to antiviral resistance[26]. Despite this observation, mutations in the nsp5 are rare[27]. We found that one of our patients who received nirmatrelvir-ritonavir (17072) developed more than one nsp5 mutation (nsp5: T169I and nsp5: A173T) reaching consensus levels (>50%). While mutations inducing decreased sensitivity to nirmatrelvir were generated in other studies after treatment in cell culture in vitro[28] and were used to identify patterns leading to resistance, data obtained directly from clinical patients are limited[29]. The nsp5 A173T, observed in patient 17072, was previously shown to cause a 4.1-fold increase in resistance to nirmatrelvir[30]. The nsp5 T169I has been shown to emerge during passage in cell culture in the presence of increasing concentrations of nirmatrelvir and was shown to decrease the sensitivity (-twofold) of variant isolates to nirmatrelvir[13,15,30]. The nsp5 T169I mutation was detected in the day 52 sample from patient 17072, representing a minor variant with a frequency of 34% and then increasing to 92% by day 77 and decreasing to 62% by day 81. The SARS-CoV-2 variant carrying the double nsp5 T169I and nsp12 V792I mutations became the predominant virus in the respiratory secretion of patient 17072 and was successfully isolated in cell culture, replicating to wild-type virus levels in vitro, demonstrating that the mutant virus had no apparent loss of replication fitness.

In vitro antiviral testing revealed that the SARS-CoV2-nsp5$^{T169I}$nsp12$^{V792I}$ virus presented a higher $IC_{50}$ (-twofold) for nirmatrelvir and remdesivir when compared to the wild-type virus. Although the presence of these mutations conferred a low-level resistance to SARS-CoV2-nsp5$^{T169I}$nsp12$^{V792I}$, the observed decreased sensitivity to antiviral treatment may delay viral clearance and complicate the outcome of SARS-CoV-2 infection in immunocompromised patients. Our results showing the inhibition of viral replication with a combination of nirmatrelvir and remdesivir treatment in vitro suggests that combination therapy with multiple antiviral drugs could provide a better treatment alternative than monotherapy, as also suggested by non-randomized clinical studies in high-risk and immunocompromised patients[31,32].

Structural and molecular docking analysis of nsp5$^{T169I}$ and nsp12$^{V792I}$ revealed that although the amino acid residues affected by these mutations do not lie within the nirmatrelvir or remdesivir binding pockets of nsp5 and nsp12, respectively, these amino acid substitutions lead to marked changes in the residues that interact with the drugs. This, in turn, is predicted to result in changes in the number of hydrogen bonds and hydrophobic interactions between the antiviral drug and amino acid residues within the protein's binding pocket suggesting a potential mechanism of action that, however, remains to be elucidated experimentally.

Our findings have important clinical implications. First, the rapid emergence of resistance driven by the administration of antiviral monotherapy observed in our study underscores the need for studies to identify the best treatment options for immunocompromised patients. This finding is also supported by another report on the acquisition of multiple resistance mutations in response to antiviral therapy in an immunocompromised patient experiencing treatment failure[33]. Second, although the prevalence of resistance mutations to remdesivir or nirmatrelvir-ritonavir of circulating SARS-CoV-2 strains is low[27], the high frequency of mutations observed in our study cohort—including two patients who developed resistance to more than one drug suggests that immunocompromised populations could function as reservoirs for the emergence of resistant strains.

Finally, recovery of infectious and transmissible SARS-CoV-2-nsp5$^{T169I}$nsp12$^{V792I}$ virus with decreased sensitivity to nirmatrelvir and remdesivir more than two months after initial diagnosis highlights the potential for dissemination of variants to the general population. Our results demonstrated that the resistant SARS-CoV-2-nsp5$^{T169I}$nsp12$^{V792I}$ virus was as transmissible as the wild-type virus in a hamster model of SARS-CoV-2 infection, underscoring the need for increased surveillance in immunocompromised patients. These results also highlight the need for enhanced infection control measures in these patient populations to prevent transmission of potentially antiviral-resistant viruses to the general population.

## Methods
### Study cohort
All procedures were performed in compliance with the Declaration of Helsinki principles and approved by the Weill Cornell Medicine Institutional Review Board (IRB). Nasopharyngeal swabs from immunocompromised and immunocompetent patients were collected following signature of an informed consent as part of a prospective, observational, single-center cohort study conducted from April 7, 2020, in patients aged 18 years or older diagnosed with COVID-19 and followed at New York Presbyterian Hospital (NYP) (IRB#20-03021645)[34–36]. Additional samples from patients with hematological malignancies included in our study were obtained from Weill Cornell Biobank (IRB # 20-03021671) or from discarded samples collected for clinical care (IRB# 1506016280). Demographics, clinical and laboratory data, including treatments for COVID-19 and for the underlying malignancies, were collected.

### Detection of anti-SARS-CoV-2 antibodies in patients
Detection of anti-SARS-CoV-2 antibodies was performed by the NYP central laboratory for patient care using the Roche Elecsys Anti-SARS-CoV-2 S assay.

### SARS-CoV-2 detection and genome sequencing
The Cobas (Roche) and Xpert Xpress SARS-CoV-2 (Cepheid) RT-PCR assays were performed for routine clinical diagnosis according to the manufacturer's instructions[37]. Viral loads were measured through surrogate markers of cycle threshold (CT) values for SARS-CoV-2-specific gene targets (ORF1ab gene for the cobas assay and N2 gene for the Xpert Xpress assay). Nasopharyngeal swabs collected from patients were sequenced using Illumina platforms. The genomic viral RNA was amplified using an in-house primer set and protocol, as previously described[38,39]. A subset of ten samples was amplified using the ARTIC V4 primer set and protocol, and sequencing libraries were prepared using the Illumina DNA Prep kit (Illumina, San Diego, CA) according to the manufacturer's protocol. Pools were sequenced on either the NextSeq500 (NextSeq 500 MID) or NextSeq2000 (P1). Amplification, library preparation, and sequencing were performed in duplicate using RNA from the same extraction. SARS-CoV-2 from six samples was sequenced once using Molecular Loop Viral RNA Target Capture Kits (Molecular Loop) following the manufacturer's recommendations. Briefly, 6 μL of RNA was reverse transcribed, and capture probes were annealed in a 16-h incubation at 55 °C. The probes were then enzymatically circularized to capture the viral genome and add Unique Molecular Indexes, followed by amplification of circularized cDNA targets using 27 cycles of PCR. Libraries were sequenced on a NovaSeq 6000 sequencer with 2× 150 bp reads. Samples collected from hamsters were sequenced using a tiled multiplexed amplicon approach on the GridION platform (Oxford Nanopore Technologies, ONT). Briefly, total RNA was reverse-transcribed and amplified as

described in the protocol available at https://doi.org/10.17504/protocols.io.br54m88w, using custom primers (Supplementary Table 10) or ARTIC V4.1 primers (IDT). Library preparation was performed using a modified ARTIC network's nCoV-2019 sequencing protocol v2 (https://www.protocols.io/view/ncov-2019-sequencing-protocol-v2-bdp7i5rn). Final sequencing libraries were loaded onto R9.4 flow cells. The output FASTQ files from the hamster samples were processed through the ARTIC ncov-2019 bioinformatic pipeline, using Medaka for variant calling (https://artic.network/ncov-2019/ncov2019-bioinformatics-sop.html).

## Molecular loop data analysis

The 5 bp unique molecular identifier (UMI) was identified on the 5′ end of each mate pair and added to read header before the UMI and 25 bp molecular inversion probes were trimmed and output into an unmapped BAM. The unmapped BAMs were converted into fastq files using Samtools (v.1.14) and aligned to the Wuhan-Hu-1 reference (NC_045512.2) using BWA-MEM (v.0.7.17)[40]. Metadata from the unmapped bam was transferred to the mapped BAM using ZipperBams (fgbio, v.2.0.2, GitHub: https://github.com/fulcrumgenomics/fgbio). The alignments were sorted using the QueryName of the reads and the SortBam function (fgbio) to add the mapping quality and position of each mate pair (SetMateInfo, fgbio) and group the read pairs by their UMI (GroupReadsByUMI, fgbio). Consensus reads were generated from the read groups (GroupReadsByUMI, fgbio) and filtered (FilterConsensusReads, fgbio) before realigning the consensus reads to the Wuhan-Hu-1 reference genome using BWA-MEM (v.0.7.17). After processing and alignment, the average read length was 118 base pairs long. The final consensus read alignments, which varied in mean read depths (11× to 1824×), was used to identify single-nucleotide variants using timo (v4, https://github.com/GhedinSGS/timo)[39]. The snakemake pipeline for the molecular loop analysis can be found at https://github.com/GhedinSGS/SARS-CoV-2_Antiviral_Resistance. Raw molecular loop sequencing data and consensus sequences are available at BioProject PRJNA1088540.

## Illumina data analysis

The alignment pipeline for Illumina data has been previously outlined and is available at: https://github.com/GhedinSGS/SARS-CoV-2_analysis[39]. In short, reads are first quality trimmed with trimmomatic (v.0.39) and aligned to the Wuhan-Hu-1 reference (NC_045512.2) using BWA-MEM[41,42]. Primer sequences are removed using iVar (v.1.3.1) before inputting the alignments into timo (v4, https://github.com/GhedinSGS/timo) to identify single-nucleotide variants[43]. Mean read depths for each ARTIC amplicon alignment ranged from 112× to 7067×, with an average read depth across the dataset of 4637×. Consensus sequences were generated using a combination of timo and GATK outputs[44]. Consensus sequences and raw sequencing data are available under the BioProject PRJNA1088540.

## Variant analyses

Alignments with at least 75% of the SARS-CoV-2 genome covered at ≥5× read depth were kept for consensus variant analyses. The SARS-CoV-2 lineage information was obtained using NextClade (https://clades.nextstrain.org) and the consensus sequence for each sample and sequencing replicate[45]. Because the Pango sub-lineage information is influenced by read coverage, we used the NextClade lineage groupings to determine lineage reference sequences from CoV-Spectrum[46]. Each sample was then compared to its respective NextClade reference to identify SARS-CoV-2 variants. Nucleotide sites that differed between the sample and its defined SARS-CoV-2 reference sequence were determined and further filtered depending on the relative frequency of the variant. Low frequency minority variants (i.e. present at frequencies <50% in the alignment) had to be present at ≥2% at nucleotide

positions with at least 200× read depth and in both sequencing replicates for the short-read amplicon datasets. Consensus variants, found at relative frequencies ≥50%, were required to be at positions with at least 5× read depth and in both sequencing replicates of the short-read amplicon sequencing datasets. The molecular loop datasets were only sequenced once but had the same frequency and read depth cutoff requirements for identifying minority and consensus variants. Deletions and insertions were not considered in the analyses. One mutation at amino acid position nsp8:125 was removed from all samples sequenced using the in-house primer sets as it appeared to be a primer-specific batch effect that was not present when using other primers. All variants ≥2% were grouped and used to calculate the number of single-nucleotide variants found in each sample collected. While variants identified at frequencies ≥98% could represent strain-specific diversity that established the infection, they were still included in the variant calculations due to incomplete sampling and several emergent mutations reaching fixation in the host during the infection. All R code and additional variant information for the samples are located on GitHub at https://github.com/GhedinSGS/SARS-CoV-2_Antiviral_Resistance.

## Isolation of SARS-CoV-2 from clinical samples

Isolation of SARS-CoV-2 from NPS of three patients—17072 (four samples), 16902 (seven samples) and 11595 (one sample) was performed in BSL3 facilities of the College of Veterinary Medicine, Cornell University. Vero E6 TMPRSS2 cells (JCRB Cell Bank, JCRB1819) were cultured in 12-well plates ($1.5 \times 10^5$ cells/mL) for 24 h. The cells were then washed once with 1 mL phosphate-buffered saline (PBS), inoculated with 500 μL of nasal swab sample and incubated at 37 °C for 1 h (adsorption). Cells were then washed once with 1 mL PBS and replenished with 1 mL complete growth media (DMEM 10% FBS) and maintained at 37 °C until 80–90% cytopathic effect (CPE) developed. Infected cells were harvested, and cell suspension was collected following freeze-thaw, and subjected to two subsequent passages. At each passage, the isolated viruses were sequenced using the GridION sequencing platform (ONT). The sequences were compared to the sequence obtained from the original clinical sample. For comparative in vitro analyses and the animal study, a recent SARS-CoV-2 isolate collected in 2021, NYI45-21 (GenBank accession: PP446157) belonging to the Omicron BA1.1 lineage was retrieved from the virus repository of the Diel Lab at Cornell University. The NYC3/18-22 isolate from patient 17072 had high nucleotide sequence homology across the genome with the NYI45-21 isolate and carried two important mutations in nsp5 (T169I) and nsp12 (V792I) proteins associated with resistance to nirmatrelvir and remdesivir antivirals, respectively[15,30]. The NYC3/18-22 isolate is named SARS-CoV-2-nsp5$^{T169I}$nsp12$^{V792I}$, while the NYI45-21 isolate is the wild-type (WT). The titers of virus stocks were determined by plaque assays and end point dilutions. A mouse monoclonal antibody (made in-house) targeting the SARS-CoV-2 N protein (SARS-CoV-2 anti-N mAb clone B61G11)[47] was used as a primary antibody in the immunofluorescence assay (IFA).

## Viral growth kinetics

Viral growth kinetics were performed using Vero E6 (ATCC® CRL-1586™) and Vero E6 TMPRSS2 (JCRB Cell Bank, JCRB1819) cells. Cells were seeded in 12-well plates ($1.2 \times 10^5$ cells/mL) for 24 h until they reached 80–90% confluence. Cells were then infected with the SARS-CoV-2-nsp5$^{T169I}$nsp12$^{V792I}$ and WT viruses and incubated at 4 °C for 1 h for virus adsorption. The inoculum was then replaced with 1 mL of complete growth media and incubated at 37 °C. Cells and supernatant were harvested at 4-, 8-, 12-, 24-, 48- and 72-h post-inoculation and stored at −80 °C. Time point 0 was an aliquot of virus inoculum stored at −80 °C as soon as inoculation was completed. Virus titers were determined in Vero E6 TMPRRSS2 cells at each time point using

end-point dilutions and the Spearman and Karber's method and expressed as $TCID_{50}.mL^{-1}$.

## Plaque phenotype

The viral plaque phenotype was determined in Vero E6 and Vero E6 TMPRSS2 cells. For this, cells ($3 \times 10^5$ cells per well) were cultured in 6-well plates for 24 h. Cells were inoculated with SARS-CoV-2-nsp5[T169I]nsp12[V792I] and WT viruses (30 plaque forming units per well) and incubated at 37 °C for 1 h. Following that, the inoculum was removed and 2 mL of media containing 2× complete growth media and 0.5% SeaKem agarose (final conc. 1× media 0.25% agarose) was added to each well. Once agarose polymerized, the plate was transferred to the incubator at 37 °C for 72 h. The agarose overlay was removed, cells were fixed with 3.7% formaldehyde solution for 30 min and stained with 0.5% crystal violet solution for 10 min at room temperature. The plaque size was quantified using a Keyence BZ-X810 Microscope.

## Antiviral resistance analysis of SARS-CoV-2 in vitro

The antiviral resistance of the SARS-CoV-2 isolates nsp5[T169I]nsp12[V792I] and WT was performed in BSL3 facilities of the College of Veterinary Medicine, Cornell University using two antiviral drugs, nirmatrelvir (HY-138687, MedChemExpress), a 3C-like protease ($3CL^{PRO}$) inhibitor, and remdesivir (HY-104077, MedChemExpress), a viral RNA-dependent RNA polymerase (RdRp) inhibitor. Vero E6 cells ($1 \times 10^4$/well) were seeded in 96-well plates for 24 h. The antivirals were applied at three different time periods of infection: before infection (pre-treatment), at infection (simultaneous treatment) and after infection (post-treatment). To obtain the 50% inhibitory concentration ($IC_{50}$) of both antivirals, eleven serial twofold dilutions of nirmatrelvir (25 μM to 0.025 μM) and remdesivir (100 μM to 0.098 μM) were prepared in DMEM 2% FBS. For pre-treatment, cells were treated with nirmatrelvir or remdesivir for 4 h before infection. For simultaneous treatment, antiviral drug was added with the virus inoculum during infection. For post-treatment, antiviral drug was added 1 h after infection. Cells were infected with 200 $TCID_{50}$/well of SARS-CoV-2-nsp5[T169I]nsp12[V792I] and WT viruses. In both pre-treatment and simultaneous treatment regimens, the antiviral drugs were added and maintained for 48 h post-infection. After 48 h of infection, the cell supernatant was collected and titrated in Vero E6 TMPRSS2 cells using end-point dilutions and the Spearman and Karber's method and expressed as $TCID_{50}.mL^{-1}$. The percent inhibition of SARS-CoV-2 by nirmatrelvir and remdesivir was calculated compared to infected untreated cells and imported into GraphPad Prism 9.0 to obtain the $IC_{50}$ values of each antiviral drug for each treatment regimen. The cell monolayer was fixed with 3.7% formaldehyde solution and stained by immunofluorescence assay (IFA) using SARS-CoV-2 N specific mouse mAb.

We tested for antiviral resistance of SARS-CoV-2-nsp5[T169I]nsp12[V792I] using a combination antiviral treatment. For this, we selected the $IC_{50}$ value of each antiviral drug obtained for each treatment regimen and prepared serial twofold dilutions below (0.5× $IC_{50}$ and 0.25× $IC_{50}$) and above (4× $IC_{50}$ and 2× $IC_{50}$) the $IC_{50}$ values. Vero E6 cells were treated with either nirmatrelvir or remdesivir alone or in combination with nirmatrelvir and remdesivir using 4×, 2×, 1×, 0.5x and 0.25× $IC_{50}$ concentrations following the three different treatment regimens described above. Cells were infected with 200 $TCID_{50}$/well of SARS-CoV-2-nsp5[T169I]nsp12[V792I] and supernatant was collected after 48 h and titrated using endpoint virus dilutions and expressed as $TCID_{50}.mL^{-1}$. The cell monolayer was fixed in 3.7% formaldehyde solution and stained by IFA for SARS-CoV-2 N detection.

## 3D homology model generation

The nsp5 and nsp12 protein structure models of the nsp5[T169I] and nsp12[V792I] proteins were built and compared with WT NYI45-21 nsp5 and nsp12 sequences using the three-dimensional structure of proteins available in the Research Collaboratory for Structural Bioinformatics Protein Data Bank (RCSB PDB). The crystal structures of nsp5 with PDB ID- 8GFU[48] and nsp12 with PDB ID- 7BV2[49] were used for the modelling of nsp5 and nsp12 proteins. The nsp5 and nsp12 protein sequences of both mutant and WT viruses were retrieved from our sequencing data.

The SWISS homology modelling tool was used to generate the models of nsp5 and nsp12 proteins considering that only 1 or 2 amino acid changes (nsp5: T169I, nsp12: V223I and V792I) were observed between the SARS-CoV-2-nsp5[T169I]nsp12[V792I] and WT viruses[50]. The crystal structures 8GFU (nsp5) and 7BV2 (nsp12) were used as templates to generate the homology model. The target sequence file and template file (.pdb) were uploaded and initiated in the online SWISS model server. The 3D structure models were developed for both mutant and WT sequences and these models were further analyzed for their structural features and validated by Ramachandran plot using the PROCHECK server tool[51]. The root mean square deviation (RMSD) of the nsp5 and nsp12 models between the mutant and WT virus sequences was calculated using the PyMOL Molecular Graphics System, Version 2.3.4, Schrodinger, LLC (PyMOL | Schrödinger)[52]. Ramachandran plot analysis of the nsp5 T169I protein model demonstrated 91.3% amino acids in the most favored region, with 7.6% and 0.4% amino acids in additional allowed regions and generously allowed regions, respectively. The nsp12 V792I protein model demonstrated 90% amino acids in the most favored region, with 8.9% and 0.7% amino acids in additional allowed region and generously allowed regions, respectively, which confirmed the overall reliability of the model. Structural alignment of nsp5 T169I and nsp12 V792I models with their respective crystal structure revealed a significant degree of structural similarity with an RMSD value of 0.05 Å and 0.09 Å, respectively.

## Molecular docking

Molecular docking was performed to characterize the interaction(s) between the drug molecules nirmatrelvir and remdesivir at catalytic active sites of nsp5 and nsp12, respectively, using the WT and nsp5[T169I] and nsp12[V792I] mutant protein models. Remdesivir functions as a product requiring conversion into its active remdesivir triphosphate (RTP) form within the host cell. This active form proficiently binds to the RdRp active site and inhibits viral replication. Hence, RTP was employed in docking studies performed here. The in silico generated nsp5 and nsp12 models were energy minimized and converted into.pdbqt files using the AutoDock Vina algorithm[53,54]. The ligands nirmatrelvir (PubChem CID: 155903259) and remdesivir triphosphate (PubChem CID: 56832906) were downloaded from the PubChem database[55] in.sdf format and saved in.pdbqt format using PyRx tools version 0.8, open-source software featuring an intuitive user interface[56].

Molecular docking utilized specific parameters, with the grid center points for nsp5 protein set at X = 74, Y = 54, and Z = 58, and box dimensions set as 10.291 Å × -2.921 Å × 17.439 Å with an exhaustiveness of 8. Grid center points for nsp12 were set at $X = 58$, $Y = 62$, and $Z = 61$, and box dimensions set as 96.584 Å × 90.942 Å × 104.616 Å with an exhaustiveness of 8. The molecular interactions between the nirmatrelvir and RTP with nsp5 and nsp12 protein models, respectively, were visualized and analyzed using LIGPLOT$^+$ [57] and PyMOL tools (PyMOL | Schrödinger), which represent visually hydrophobic and hydrogen (H) bond interactions in protein–ligand interactions.

## Transmission study in golden Syrian hamsters

**Animal housing and experimental design.** A total of sixteen (8 male, 8 female) 50-day-old LVG golden Syrian hamsters (strain 049) were purchased from Charles River (United States). The average body weight of the hamsters was 94.5 g (range 83–102 g). All animals were housed in the animal biosafety level 3 (ABSL-3) facility at the East Campus Research Facility (ECRF) at Cornell University. On day 0, hamsters were inoculated intranasally with 100 μL of virus suspension

carrying $5 \times 10^4$ PFU of SARS-CoV-2-nsp5$^{T169I}$nsp12$^{V792I}$ or WT viruses ($n = 4$/virus, 2 male and 2 female) and were housed individually in cages. On the next day, each of the infected hamsters was transferred to the isolator containing the contact animal and co-housed for 13 days. Animals were monitored daily for clinical signs and body weight gain for 2 weeks. Oropharyngeal swabs were collected daily from day 0–8 and on day 10 and 14 post-inoculation (pi) or post-contact (pc). Upon collection, swabs were placed in sterile tubes containing 1 mL of viral transport medium (VTM Corning®, Glendale, AZ, USA) and stored at -80 °C until processed for further analyses. Blood samples were collected on days 0 and 14 pi and sera were separated. All hamsters were humanely euthanized on day 14 pi and nasal turbinate, trachea and lungs were harvested for virological investigation. The study procedures were reviewed and approved by the Institutional Animal Care and Use Committee at Cornell University (IACUC approval number 2021-0021).

### Nucleic acid isolation and real-time reverse transcriptase PCR (rRT-PCR)

Nucleic acid was extracted from oropharyngeal swabs and tissues collected at necropsy. A 10% (w/v) homogenate was prepared in DMEM from tissues (nasal turbinate, trachea, and lungs) using a stomacher (one speed cycle of 60 s, Stomacher® 80 Biomaster). The tissue homogenate was clarified by centrifuging at 2000×$g$ for 10 min. In total, 200 μL of oropharyngeal swabs and clarified tissue homogenate was used for RNA extraction using the MagMax Core extraction kit (Thermo Fisher, Waltham, MA, USA) and the automated KingFisher Flex nucleic acid extractor (Thermo Fisher, Waltham, MA, USA). The rRT-PCR for total viral RNA detection was performed using the EZ-SARS-CoV-2 Real-Time RT-PCR assay (Tetracore Inc., Rockville, MD, USA), which detects both genomic and subgenomic viral RNA targeting the viral nucleoprotein gene. An internal inhibition control was included in all reactions. Positive and negative amplification controls were run side-by-side with test samples. Relative viral genome copy numbers were calculated based on the standard curve and determined using GraphPad Prism 9 (GraphPad, La Jolla, CA, USA). The amount of viral RNA detected in samples was expressed as log (genome copy number) per mL.

### Virus isolation and titration

All oropharyngeal swabs and tissue homogenates were subjected to virus isolation under Biosafety Level 3 (BSL-3) conditions at the Animal Health Diagnostic Center (ADHC) Research Suite at Cornell University. The virus isolation was performed in Vero E6 TMPRSS2 and three blind passages were performed. For virus titration, serial 10-fold dilutions of samples were prepared in DMEM and inoculated into Vero E6 TMPRSS2 cells in 96-well plates. Two days later, culture supernatant was aspirated, and cells were fixed with 3.7% formaldehyde solution and subjected to IFA as described previously[47]. The limit of detection (LOD) for infectious virus titration is $10^{1.05}$ TCID$_{50}$.mL$^{-1}$. Virus titers were determined on each time point using end-point dilutions and the Spearman and Karber's method and expressed as TCID$_{50}$.mL$^{-1}$.

### Neutralizing antibodies

Neutralizing antibodies in serum against SARS-CoV-2 was assessed by virus neutralization (VN) assay. Serum samples collected on days 0 and 14 pi were tested against SARS-CoV-2-nsp5$^{T169I}$nsp12$^{V792I}$ virus. To this end, serial twofold serum dilutions (1:8 to 1:1024) were incubated with 100–200 TCID$_{50}$ of the SARS-CoV-2 nsp5$^{T169I}$nsp12$^{V792I}$ isolate for 1 h at 37 °C. Following that, 50 μL of a cell suspension of Vero E6 cells was added to each well of a 96-well plate and incubated at 37 °C. Two days later, cells were fixed and subjected to IFA as described in a previous study[47]. Neutralizing antibody titers were expressed as the reciprocal of the highest dilution of serum that completely inhibited SARS-CoV-2

infection/replication. FBS and positive and negative serum samples from cat were used as controls.

### Statistical analysis

Statistical analysis was performed by two-way analysis of variance (ANOVA) followed by multiple comparisons. The Mann–Whitney $U$ test was used to compare plaque sizes between the resistant and wild-type viruses. Statistical analysis and data plotting were performed using the GraphPad Prism software (version 9.0.1).

### Reporting summary

Further information on research design is available in the Nature Portfolio Reporting Summary linked to this article.

## Data availability

Additional clinical data for the patients presented in this article are not readily available due to protection of individual privacy. Requests to access additional data should be directed to: mis2053@med.cornell.edu. Consensus and raw sequencing data for all patient samples are available at BioProject PRJNA1088540. Source data are provided with this paper.

## Code availability

Code for aligning, processing, and variant calling can be found at https://github.com/GhedinSGS/SARS-CoV-2. Code for variant analyses are available at https://doi.org/10.5281/zenodo.13306528 (v1) and https://github.com/GhedinSGS/SARS-CoV-2_Antiviral_Resistance.

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

## Acknowledgements

We thank the Center for Animal Resources and Education (CARE) staff, Cornell Biosafety team, Weill Cornell Medicine biorepository staff and New York Genome Center (NYGC) Sequencing Laboratory staff for their support. Sequencing was performed in part by the NYGC Sequencing Laboratory as part of the COVID-19 Genomic Research Network (CGRN) with funds generously provided by NYGC donors and the JPB Foundation. This work was funded in part by the National Institutes of Health (NIH) and National Institute of Allergy and Infectious Diseases (NIAID) grant no. R01AI166791-01 (to D.G.D.), by the Division of Intramural Research of the NIAID/NIH (E.G.), by the Office of Vice President for Research at Cornell University and the Office of the Dean for Research and Graduate Education at the College of Veterinary Medicine at Cornell University (D.G.D.) and by the Weill Cornell Medicine National Center for Advancing Translational Science of the National Institute of Health (award number UL1TR002384). This work used the computational resources of the NIH High Performance Computing (HPC) Biowulf cluster (http://hpc.nih.gov) and the Office of Cyber Infrastructure and Computational Biology (OCICB) HPC cluster at NIAID, Bethesda, MD.

## Author contributions

Conceptualization: M.S., E.G. and D.G.D.; methodology: M.N., K.E.E.J., W.W., S.B. and J.A., L.C.C.; software: M.N., K.E.E.J., R.R. and L.C.C.; formal analysis: M.N., K.E.E.J., R.R., E.J.F., L.C.C., M.S., E.G. and D.G.D.; investigation: M.N., K.E.E.J., R.R., E.J.F., L.C.C., W.W., S.B., J.A., R.P.K., J.H., M.T.S., L.W., C.Z., L.M. and R.S.; data curation: K.E.E.J., M.N., E.J.F., R.R. and L.C.C.; writing—original draft: M.N., K.E.E.J., E.G., D.G.D. and M.S.; writing—review and editing: all authors; visualization: M.N., K.E.E.J., R.R. and E.J.F.; supervision: E.G., D.G.D. and M.S.; project administration: E.G., D.G.D. and M.S.; funding acquisition: E.G. and D.G.D.

## Competing interests

The authors declare no competing interests.
