## [Peer Review File · Nature Communications]

Emergence of transmissible SARS-CoV-2 variants with decreased sensitivity to antivirals in immunocompromised patients with persistent infectionsReviewers' Comments:

Reviewer #1:

Remarks to the Author:

In this study, the authors investigated the amino acid mutations at nsp5 and nsp12 in the antiviral-treated patients suffering from persistent SARS-CoV-2 infection due to immunosuppression. Among several amino acid mutations detected, they focused on nsp5-T169I and nsp12-V792I substitutions. SARS-CoV-2 carrying both substitutions showed a 2-3-fold reduction in susceptibility to nirmatrelvir and remdesivir. In addition, the mutant virus was transmitted between hamsters by direct contact.

Overall, most of the data are solid and reasonable for the antiviral study. However, several key data are missing, as described below.

1. The authors need to confirm the resistance of SARS-CoV-2 with nsp5-T169I and nsp12-V792I substitutions to monotherapy with nirmatrelvir or remdesivir and to combination therapy with nirmatrelvir and remdesivir in the hamster infection model.
2. The authors need to test the airborne transmissibility of SARS-CoV-2 with nsp5-T169I and nsp12-V792I substitutions between hamsters.

Reviewer #2:

Remarks to the Author:

In this manuscript, Nooruzzaman et al., explored the resistance emergence profile in immunodeficient patients treated with Nirmatrelvir, remdesivir or both. First of all, I would like to thank the authors for this very important study and for the important findings. Overall, the manuscript is very well written and the data representation is very clear leading to solid conclusions. I have only minor comments:

1. The EC50 of Remdesivir and nirmatrelvir in the displayed in vitro data against WT virus is higher than usually reported for both compounds in Vero cells, is there an explanation for that?
2. did you check for stability of the nsP5 and nsP12 mutations of the double resistant mutant by passaging in vitro and/or in vivo?

Reviewer #3:

Remarks to the Author:

The manuscript by Nooruzzaman, Johnson, and collaborators describes the emergence of antiviral-resistant mutations in three SARS-CoV-2 strains from 15 persistent infections in immunocompromised individuals. Currently, there is no standard treatment to ensure viral

clearance of persistent SARS-CoV-2 infection to prevent the emergence of novel variants, and this study provides relevant clinical and experimental data to understand the impact of antiviral treatment.

The manuscript is well-written and easy to follow, and the analyses and experiments are well done. The experimental data with combination therapy are very promising. I'm not a clinician, but the authors could mention whether simultaneous treatment with Paxlovid and Remdesivir is clinically feasible (in terms of tolerability) for patients undergoing chemotherapy or other immunosuppressive therapy.

Regarding the genomic analysis, the authors do not mention whether related mutations to the other treatments (i.e. monoclonal antibodies) were observed in these patients. The authors must have looked at other regions, so please consider including a brief description of other genomic changes observed before directing your focus to nps5 and nps12 alone. A supplementary table summarizing this information could be included. This is a valuable resource to add to the cumulative evidence of the importance of intrahost evolution in the emergence of SARS-CoV-2 variants. Equally important is to mention if no other mutations were observed.

A few suggestions on the text:

Abstract:

Line 25, please include the number of patients that received each treatment rather than "some."

Results:

Line 77: For clarity, I suggest adding to the sentence "Nsp5 mutations were identified in 4 of the 15 patients..." that this included one of the patients who received Paxlovid before describing the nps12 mutations. Also, include this information in Supplementary Table 2 to make it easier for the reader to identify the cases of interest.

Line 100 and Fig. 1A. Please indicate lineages and clades consistently across the figures and text. Since you have repeated samples, including the PANGO lineage assigned to the earliest sample would be sufficient.

Line 106, "Four [other] nps12 substitutions were identified..." refers to substitutions that have also been reported to confer remdesivir resistance. If so, this is not fully clear, but references are included. Consider rewording this statement to clarify.

Line 108. "Were detected in the first sample collected" add "and sequenced." Figure 1A also shows the collected sample but not sequenced.

Lines 163 and 168, replace lineage "B1" for B.1. which is the proper nomenclature.

Methods:

Line 343: What is NYP? Please define it. Also, ensure it is consistent with line 349, where New York Presbyterian is mentioned.

Sequencing sections: Line 362, were all samples sequenced by both sequencing methods? Please be specific. What is the molecular loop amplicon length? Does this method allow variant phasing? If so, was this information used to confirm that both nps5/nps12 mutations were present in the same genome?

Also, it is unclear if cultured viral stocks were plaque-purified or free of other iSNVs.

The authors included a BioProject identifier for sequencing data, which was not yet accessible

during this review. Does this include consensus and raw sequencing reads? Both data sets should be made available.

Line 400, thank you got the clear description of the variant analysis. What was the average read depth per genome?

Figure 1A. Indicate the Pango lineage of the earliest or highest confidence sample.

Figure 1A-D. What was the reference genome used for each panel? Was it the prototype strain of each detected lineage or ancestral Wuhan01? Also, it needs to be clarified if the number of SNVs shown in the top chart of each panel corresponds to the whole genome or only nsp5/nsp12.

Consider splitting panel D into two panels, one for nsp5 and nsp12, respectively.

Figure 2B-C. I suggest decreasing the alpha for the plotted data points to make all overlapping dots visible.

Please find point-by-point responses outlined in blue, to reviewer comments below.

Reviewer #1:

-
1. The authors need to confirm the resistance of SARS-CoV-2 with nsp5-T169I and nsp12-V792I substitutions to monotherapy with nirmatrelvir or remdesivir and to combination therapy with nirmatrelvir and remdesivir in the hamster infection model.
 2. The authors need to test the airborne transmissibility of SARS-CoV-2 with nsp5-T169I and nsp12-V792I substitutions between hamsters.

While we agree the experiments proposed by the reviewer are of high interest, we believe they are beyond the scope of the current study.

Reviewer #2:

The EC50 of Remdesivir and nirmatrelvir in the displayed in vitro data against WT virus is higher than usually reported for both compounds in Vero cells, is there an explanation for that?

We agree with the reviewer that the IC₅₀ values obtained for remdesivir against the WT Omicron BA.1.1 virus were higher than ones observed in other published studies. The differences could be attributed to a relatively higher replication of the virus in Vero cells producing up to 6.8 Log₁₀ TCID₅₀/mL titer in 48 hours (**Supplemental Figure 2B**). IC₅₀ values were calculated based on virus titers of the cell culture supernatant, which relied on a highly sensitive immunofluorescence assay.

Did you check for stability of the nsP5 and nsP12 mutations of the double resistant mutant by passaging in vitro and/or in vivo?

We performed three serial passages of the double resistant mutant in Vero E6 TMPRSS2 cells and both nsp5 T169I and nsp12 V792I mutations were found to be stable in all 3 passages. We did not pass the double mutant *in vivo*. However, whole genome sequencing of SARS-CoV-2 from oropharyngeal swab samples collected between days 1 and 7 from SARS-CoV-2-nsp5^{T169I}nsp12^{V792I} inoculated and contact hamsters showed that both nsp5^{T169I} and nsp12^{V792I} mutations were maintained in the virus upon replication and transmission in hamsters.

Reviewer #3:

(...) The experimental data with combination therapy are very promising. I'm not a clinician, but the authors could mention whether simultaneous treatment with Paxlovid and Remdesivir is clinically feasible (in terms of tolerability) for patients undergoing chemotherapy or other immunosuppressive therapy.

The following sentence has been added in the discussion: **“Combination therapy with multiple antiviral drugs could provide a better treatment alternative than monotherapy, as also suggested by non-randomized clinical studies in high-risk and immunocompromised patients.”**

Regarding the genomic analysis, the authors do not mention whether related mutations to the other treatments (i.e. monoclonal antibodies) were observed in these patients. The authors must have looked at other regions, so please consider including a brief description of other genomic changes observed before directing your focus to nsp5 and nsp12 alone. A supplementary table summarizing this information could be included. This is a valuable resource to add to the cumulative evidence of the importance of intrahost evolution in the emergence of SARS-CoV-2 variants. Equally important is to mention if no other mutations were observed.

We agree with the reviewer that mutations outside of nsp5 and nsp12 are also important to consider. We have included a supplemental figure (**Supplementary Fig. 1**) outlining non-synonymous mutations that are present at multiple timepoints throughout infection in patients 11595, 16902, and 17072. In the text, we highlight the mutations that may be linked to the nsp5/nsp12 mutations of interest, become dominant, or are spike mutations that may impact antibody binding affinity or confer Sotrovimab resistance.

A few suggestions on the text:

Abstract:

Line 25, please include the number of patients that received each treatment rather than “some.”

The abstract text is adjusted to include the number of patients in our cohort who were also treated with nirmatrelvir-ritonavir (n=3) or monoclonal antibodies (n=4). The text now reads:

Line 25: “All patients received remdesivir and some also received nirmatrelvir-ritonavir (n= 3) or therapeutic monoclonal antibodies (n=4).”

Results:

Line 77: For clarity, I suggest adding to the sentence “Nsp5 mutations were identified in 4 of the 15 patients...” that this included one of the patients who received Paxlovid before describing the nsp12 mutations. Also, include this information in Supplementary Table 2 to make it easier for the reader to identify the cases of interest.

We adjusted the text to include the number of patients who received nirmatrelvir-ritonavir and have included treatment information in Supplementary Table 2. The text now reads:

Line 77: “Nsp5 mutations were identified in 4 of the 15 patients, including one patient who received nirmatrelvir-ritonavir before samples were collected and sequenced and one who was treated with nirmatrelvir-ritonavir following sample collection. In comparison, nsp12 mutations were identified in 9 of the 15 patients (Supplementary Table 2).”

Line 100 and Fig. 1A. Please indicate lineages and clades consistently across the figures and text. Since you have repeated samples, including the PANGO lineage assigned to the earliest sample would be sufficient.

PANGO lineage information is now included in **Figure 1 and Supplementary Fig. 1, Supplementary Tables 1 and 2**, and updated throughout the text.

Line 106, “Four [other] nsp12 substitutions were identified...” refers to substitutions that have also been reported to confer remdesivir resistance. If so, this is not fully clear, but references are included. Consider rewording this statement to clarify.

Line 106 was reworded to indicate that the three mutations were identified in other remdesivir studies. The text now reads:

Line 111: “Four nsp12 substitutions were identified at single timepoints with frequencies <30%, three of which (E136A, V166L, C799F) were detected in the first sequenced sample (day 14 pd) and have been identified in studies focused on the emergence of remdesivir resistant mutations. Four mutations were present at multiple timepoints outside the nsp5 and nsp12 coding regions but never became dominant (Supplementary Fig. 1A).”

Line 108. “Were detected in the first sample collected” add “and sequenced.” Figure 1A also shows the collected sample but not sequenced.

The text now clarifies that the four nsp12 substitutions were identified in the first sequenced sample (**line 112**). The updated text is provided in the above comment.

Lines 163 and 168, replace lineage “B1” for B.1. which is the proper nomenclature.

We have adjusted the text on **lines 177 and 182** to represent the correct nomenclature as “B.1”.

Methods:

Line 343: What is NYP? Please define it. Also, ensure it is consistent with **line 349**, where New York Presbyterian is mentioned.

We clarified what NYP refers to (**lines 360-366**) and adjusted our reference in **line 360**.

Sequencing sections: Line 362, were all samples sequenced by both sequencing methods? Please be specific. What is the molecular loop amplicon length? Does this method allow variant phasing? If so, was this information used to confirm that both nps5/nsp12 mutations were present in the same genome?

We adjusted the methods section to clarify that 10 samples were sequenced with the ARTIC V4 primer set, and six samples were sequenced using the Molecular Loop method. The text now reads:

Line 374: “A subset of ten samples was amplified using the ARTIC V4 primer set and protocol, and sequencing libraries were prepared using the Illumina DNA Prep kit (Illumina, San Diego, CA) according to the manufacturer’s protocol.”

Line 379: “SARS-CoV-2 from six samples was sequenced once using Molecular Loop Viral RNA Target Capture Kits (Molecular Loop) following the manufacturer's recommendations.”

Also, it is unclear if cultured viral stocks were plaque-purified or free of other iSNVs.

We did not plaque purify the viral stocks. The stock viral sequences were checked for the iSNVs and there were 3 and 5 low frequency mutations detected in the WT and double mutant virus, respectively, which probably have no biological significance.

The authors included a BioProject identifier for sequencing data, which was not yet accessible during this review. Does this include consensus and raw sequencing reads? Both data sets should be made available.

We clarified (lines 408, 417, and 613) that both consensus sequences and raw sequencing data are available under the BioProject provided. Additionally, we moved the data availability section above the acknowledgments.

Line 400, thank you got the clear description of the variant analysis. What was the average read depth per genome?

We have included the range of mean read depths for the Molecular Loop and ARTIC-Illumina data in the methods sections. The text now reads:

Line 404-406: “After processing and alignment, the average read length was 118 base pairs long. The final consensus read alignments, which varied in mean read depths (11X to 1824X).”

Line 415-417: “Mean read depths for each ARTIC amplicon alignment ranged from 112X to 7,067X, with an average read depth across the dataset of 4,637X.”

Figure 1A. Indicate the Pango lineage of the earliest or highest confidence sample.

PANGO lineage information has been added to supplemental tables 1 and 2, updated in the text, and provided in Figure 1 and Extended Data Figure 1.

Figure 1A-D. What was the reference genome used for each panel? Was it the prototype strain of each detected lineage or ancestral Wuhan01? Also, it needs to be clarified if the number of SNVs shown in the top chart of each panel corresponds to the whole genome or only nsp5/nsp12. Consider splitting panel D into two panels, one for nsp5 and nsp12, respectively.

We have updated the legend for Figure 1 to clarify that the top panel of Figures 1B-D shows the total number of SNVs $\geq 2\%$ across the entire genome and that SNVs are determined by comparing each sample to its respective SARS-CoV-2 clade consensus sequence (with more details in the methods section). In Fig. 1D, we included a small horizontal line to visually split up the nsp5 and nsp12 mutations.

Figure 2B-C. I suggest decreasing the alpha for the plotted data points to make all overlapping dots visible

We have updated Figure 2B-C to make all overlapping dots visible.